# Covid-19: The Rollercoaster of Fibrin(Ogen), D-Dimer, Von Willebrand Factor, P-Selectin and Their Interactions with Endothelial Cells, Platelets and Erythrocytes

**DOI:** 10.3390/ijms21145168

**Published:** 2020-07-21

**Authors:** Corlia Grobler, Siphosethu C. Maphumulo, L. Mireille Grobbelaar, Jhade C. Bredenkamp, Gert J. Laubscher, Petrus J. Lourens, Janami Steenkamp, Douglas B. Kell, Etheresia Pretorius

**Affiliations:** 1Department of Physiological Sciences, Faculty of Science, Stellenbosch University, Stellenbosch 7602, South Africa; 21069921@sun.ac.za (C.G.); 20825919@sun.ac.za (S.C.M.); 21074682@sun.ac.za (L.M.G.); 20006721@sun.ac.za (J.C.B.); 2Elsie du Toit Street, Stellenbosch MediClinic, Stellenbosch 7600, South Africa; laubscher911@gmail.com (G.J.L.); wodie22@icloud.com (P.J.L.); 3PathCare Laboratories, PathCare Business Centre, Neels Bothma Street, N1 City, Cape Town 7460, South Africa; janami.steenkamp@pathcare.org; 4Department of Biochemistry and Systems Biology, Institute of Systems, Molecular and Integrative Biology, Faculty of Health and Life Sciences, University of Liverpool, Crown St, Liverpool L69 7ZB, UK; 5The Novo Nordisk Foundation Centre for Biosustainability, Building 220, Kemitorve Technical University of Denmark, 2800 Kongens Lyngby, Denmark

**Keywords:** COVID-19, fibrin(ogen), thrombosis, bleeding

## Abstract

Severe acute respiratory syndrome coronavirus 2 (SARS-Cov-2), also known as coronavirus disease 2019 (COVID-19)-induced infection, is strongly associated with various coagulopathies that may result in either bleeding and thrombocytopenia or hypercoagulation and thrombosis. Thrombotic and bleeding or thrombotic pathologies are significant accompaniments to acute respiratory syndrome and lung complications in COVID-19. Thrombotic events and bleeding often occur in subjects with weak constitutions, multiple risk factors and comorbidities. Of particular interest are the various circulating inflammatory coagulation biomarkers involved directly in clotting, with specific focus on fibrin(ogen), D-dimer, P-selectin and von Willebrand Factor (VWF). Central to the activity of these biomarkers are their receptors and signalling pathways on endothelial cells, platelets and erythrocytes. In this review, we discuss vascular implications of COVID-19 and relate this to circulating biomarker, endothelial, erythrocyte and platelet dysfunction. During the progression of the disease, these markers may either be within healthy levels, upregulated or eventually depleted. Most significant is that patients need to be treated early in the disease progression, when high levels of VWF, P-selectin and fibrinogen are present, with normal or slightly increased levels of D-dimer (however, D-dimer levels will rapidly increase as the disease progresses). Progression to VWF and fibrinogen depletion with high D-dimer levels and even higher P-selectin levels, followed by the cytokine storm, will be indicative of a poor prognosis. We conclude by looking at point-of-care devices and methodologies in COVID-19 management and suggest that a personalized medicine approach should be considered in the treatment of patients.

## 1. Introduction

Severe acute respiratory syndrome coronavirus 2 (SARS-Cov-2), also known as coronavirus disease 2019 (COVID-19)-induced infection, is strongly associated with various coagulopathies [1,2,3,4,5]. Pathology might also be consistent with infection-induced inflammatory changes as observed in patients with disseminated intravascular coagulopathy [6]. Because of limited clinical patient data and the current lack of clinical trial data, it is important to investigate all possible adjuvant therapies that may contribute to a better patient outcome, specifically with regards to the coagulation profile. Of particular interest are the various circulating inflammatory coagulation biomarkers involved directly in clotting, with specific focus on fibrin(ogen), D-dimer, P-selectin and von Willebrand Factor (VWF). Changes in their levels can lead to an imbalance between procoagulant and anticoagulant factors, e.g., fibrinogen contributes to thrombus formation, while loss of high molecular VWF causes bleeding tendency [7]. During COVID-19 pathology, depending on the severity of the condition, dysregulation has been noted in all of the mentioned biomarkers, where increased levels of P-selectin [8], fibrinogen and D-dimer accompany COVID-19 progression [4,5,9,10,11,12,13,14,15] and VWFs [16,17] have been found. D-dimers are normally not present in blood unless coagulation has occurred, and the D-dimer therefore serves as a biomarker for thrombosis [18,19]. The normal range of D-dimers is <0.50 μg/mL Fibrinogen Equivalent Units (FEU). Interestingly, during early onset of the condition, D-dimer levels are normal to slightly increased (clinical observation by co-author Laubscher).

Although both low and high levels of fibrinogen (normal levels are between 2–4 mg/mL) have been reported in COVID-19 [4], central to the presence of high levels of fibrinogen is the presence of increased blood viscosity. COVID-19-associated hyperviscosity has been reported, and its presence has been ascribed to a potentially severe consequence of infection [20]. An important marker of COVID-19 disease severity might thus also be erythrocyte sedimentation rate (ESR) [21]. However, it should be noted that most patients with comorbidities will have an elevated ESR. It can be expected that, where fibrinogen levels are increased, there will be greater tendency for rouleaux formation and thus a raised ESR, as the relatively dense rouleaux will sink to the bottom faster [22]. Rouleaux formation may also happen due to the presence of inflammation and an increase in acute phase proteins in circulation [22]. Al-Samkari and co-workers reported that ESR levels of >40 mm/h (adjusted OR, 2.64 (1.07–6.51)) [4] as well as increased levels of CRP, fibrinogen, ferritin and procalcitonin could be found in patients with thrombotic complications but that the opposite could also occur in a bimodal phenomenon, where pathologically lower levels may lead to bleeding. The authors also mentioned that clinically relevant thrombocytopenia and reduced fibrinogen were rare but, when present, were associated with significant bleeding manifestations [4]. An important clinical dilemma in this pandemic is that the patient cohort is extremely diverse, with no two patients with the same clinical profile. A one-treatment-for-all regime is therefore not a useful clinical approach (and may be dangerous). Rather, a personalized patient-orientated clinical approach should be followed, where various biomarker analysis, including clotting profile point-of-care analysis, would be the most successful approach. However, there is a fine balance between time, number of patients and viable options during this pandemic. Here, we focus on some of the important biomarkers in coagulation pathology.

Fibrinogen, D-dimer, VWF and P-selectin are central in the development of coagulopathies, and coagulopathies with diverse aetiologies have been described in COVID-19 patients. An example is the augmented risk of venous thromboembolism [7]. In a cohort study involving 201 patients with confirmed COVID-19 pneumonia, risk factors associated with the development of acute respiratory distress syndrome (ARDS) and progression from ARDS to death included, among others. coagulation dysfunction [23]. For patients with ARDS who died, coagulation function indices including D-dimer (*p* = 0.001) were significantly elevated compared with patients with ARDS who survived; elevated D-dimers were prognostic of worse outcome in other reports as well [23]. As mentioned previously, during early onset of the condition, D-dimer is within normal ranges or only slightly increased but, as the patient disease severity progresses, D-dimer levels are significantly increased (clinical observation by co-author). It was also suggested that the early evaluation and continued monitoring of D-dimer levels after hospitalization may identify patients with cardiac injury and may predict further COVID-19 complications [24]. Disseminated intravascular coagulation (DIC) in COVID-19 was also found to be accompanied by a significant decrease of fibrinogen and a marked increase of fibrin(ogen) degradation product (FDP) formation and D-dimer [25]. Increased FDP and D-dimer are characteristics of DIC with hyperfibrinolysis, whereas the DIC caused by infection is accompanied by plasminogen activator inhibitor-1 release and suppression of fibrinolysis [25].

Furthermore, it was noted that, D-dimer levels on admission with fourfold increases could effectively predict in-hospital mortality in patients with Covid-19 [26] (the normal range of D-dimers is <0.50 μg/mL). One dilemma in using D-dimer or fibrinogen levels as biomarkers was pointed out by Favaloro and Thackil in 2020 [14]. The authors argue that care should be taken with regards to the units and to use standardized D-dimer and fibrinogen assays when looking at COVID-19 patient data [14]. Recent recommendations also suggest that all hospitalized COVID-19 patients should receive thromboprophylaxis or full therapeutic-intensity anticoagulation if such an indication is present [27]. However, we argue here that detailed measurement of levels of fibrin(ogen), D-dimer and other markers of hypercoagulation, especially P-selectin and VWF, are of importance during thromboprophylaxis.

Platelet levels are typically in the range 150,000 to 450,000 platelets per μL. A platelet count of less than 150,000 platelets per μL is lower than normal, and if it is below normal, the patient has thrombocytopenia. Importantly, in the context of COVID-19, the risk for serious bleeding occurs when the levels are as low as 10,000 or 20,000 platelets per μL. Together with fibrin(ogen) and D-dimer analysis, thrombocytopenia in COVID-19 is also well recognized [28,29,30,31]. Low platelet count is associated with increased risk of severe disease and mortality in patients with COVID-19 and, thus, should serve as a clinical indicator of worsening illness during hospitalization [28]. Thrombocytopenia is a well-known pathology during viral (and bacterial) infections [32]. One cause of depleted platelet numbers might be because of an increase in circulating biomarkers (including fibrin(ogen), D-dimer, P-selectin and VWF) that may directly bind to platelet receptors, followed by platelet hyperactivation and aggregation. During such hyperactivation, platelet count is lower, as hyperactivated and aggregated platelets are not counted during a platelet count analysis. During COVID-19 pathogenesis, endothelial activation and interaction with the various inflammatory biomarkers as well as the virus material may also be crucial.

In this paper, we discuss the nexus between COVID-19 and circulating inflammatory biomarkers, with a particular focus on fibrin(ogen), its breakdown products (especially D-dimer), P-selectin and VWF. We review the literature that shows how circulating biomarkers could be used in the early detection of risk of increased disease severity and argue that they should therefore be helpful markers to improve the management of COVID-19 patients. We first discuss how fibrin(ogen) D-dimer, VWF and P-selectin interact with platelets, endothelial cells and erythrocytes. We then propose a mechanism regarding how these biomarkers and, particularly, fibrin(ogen) may be involved in COVID-19 hypercoagulation and thrombocytopenia. See Figure 1 for a general layout of this review. Figure 2A shows the typical pathology in bleeding and clotting, while Figure 2B shows the fine balance between these biomarkers and the development of hyperclotting and thrombosis followed by thrombocytopenia, bleeding and the cytokine storm during COVID-19.

## 2. Discussion

### 2.1. The Importance of Fibrin(Ogen) and Its Breakdown Product, D-Dimer, as Circulating Biomarkers

Figure 3 shows the structure of soluble fibrinogen and how it polymerizes into insoluble fibrin fibre (blood) clots under the action of thrombin. Fibrinogen is a large, extracellular protein, synthesised by the liver and mainly found in the blood [33,34,35]. Fibrinogen is also an acute-phase protein that is upregulated during inflammation [33,36,37]. Upregulation of fibrin(ogen) is associated with hypercoagulability and endothelial dysfunction. Increased fibrinogen levels are also associated with many inflammatory conditions including hypertension, diabetes and thrombotic strokes [33,34,38,39,40]. Blood clots are dissolved by plasmin, a protein which degrades fibrin networks, producing fibrin degradation products [33], which include D-dimer [41]. D-dimer is also an important circulating inflammatory biomarker [42,43,44,45,46]. The D-dimer protein contains two cross-linked D fragments from the fibrinogen protein formed upon degradation of the fibrin gel, the core component of blood clots [18]. Fragment D consists of all three (α-, β- and γ-) chains that are components of intact fibrinogen [47].

Enhanced fibrin synthesis activates plasminogen and the resulting plasmin cleaves the fibrin network into soluble fragments [41]. Plasmin cleavage between the D and *E* domains yields (DD)E, the noncovalent complex of D-dimer (DD) and fragment E. Further proteolysis liberates fragment E from DD [41]. Lysis of crosslinked fibrin by plasmin therefore produces D-dimer-containing γ-γ crosslinks that hold 2 D-regions together [48]. In short, the morphology of D-dimer is characterized by the cross-linked, cleaved, identical monomers which each serve as D-dimer domains (see Figure 3). We also recognize the widespread prevalence in infection-related disease of an abnormal pathway of blood clotting to create an amyloid form of fibrin that is highly resistant to degradation [49,50,51,52,53,54,55,56,57,58,59,60,61,62,63,64]. Assessing this in COVID-19 patients would seem to be an important direction.

#### 2.1.1. Interaction of Fibrin(Ogen) and D-Dimer with Cellular Receptors

Although coagulation is the primary function of fibrinogen, it also interacts with other plasma components such as platelets, endothelial cells (ECs), erythrocytes and extracellular proteins [34]. Fibrin(ogen) receptors are of particular importance, as binding of their ligands causes the activation of various inflammatory signalling pathways. These pathways are important in healthy physiological processes but play crucial roles in pathophysiology, including the cytokine storm in COVID-19. Poor outcomes in COVID-19 correlate with clinical and laboratory features of cytokine storm syndrome [65] and increased D-dimer levels. However, when the cytokine storm is present in a patient, bleeding is prevalent and a low survival rate is then noted (clinical observation). See Table 1 for fibrin(ogen) platelet receptors.

#### 2.1.2. Fibrin(Ogen) and D-Dimer Receptors and Pathways in Platelets

Platelets contain three types of granules—α-granules, dense granules and lysosomes [86]. Ligand binding, including fibrin(ogen) and D-dimer to platelet receptors, followed by the activation of signalling pathways, leads to the secretion of molecules stored in these granules. Granule secretion results in platelet activation, aggregation and thrombus growth.

Soluble fibrinogen mainly binds to integrins on platelets, activating the platelets and promoting the formation of a platelet clot [40,86,87]. Integrins are cell-surface transmembrane receptors responsible for platelet aggregation and adhesion of cells to vessel walls [79,86,87]. Integrin αIIbβ3 signalling is one of the important platelet processes where fibrinogen binding is involved [80] and is involved in platelet spreading [88] (see Figure 4 for the integrin αIIbβ3 structure). Other integrins to which fibrinogen binds have also been identified, not only on platelets but also on endothelial cells; see Table 1 for such receptors.

Integrin αIIbβ3 receptors in the membranes of platelets are usually inactive, resulting in a low affinity for ligands [79]. Activation of a platelet by other ligand-receptor binding events can convert integrins to a higher affinity state, resulting in conformational changes to the receptors, and allows for additional signalling events [40]. Such a signalling event is called “inside-out” signalling/activation [79]. When platelets are activated due to binding of biomarkers to a membrane receptor, inside-out signalling pathways increase the affinity of αIIbβ3 for fibrinogen (and other ligands). The binding of biomarkers to active integrin αIIbβ3 receptors then results in outside-in signalling. An example of this process is where αIIbβ3 receptor binding is then dependent on FcγRIIa ITAM (immunoreceptor tyrosine-based activation motif)/spleen tyrosine kinase (Syk)/PLCγ2 and phosphatidylinositide 3-kinase (PI3K)/Akt to amplify the platelet activation [40,92,93] This integrin receptor activation is in many cases caused by receptor clustering and where the integrins form complexes (or heteroclusters) with other receptors (such as GPIb, glycoprotein VI (GPVI) or FcγRIIa) [79].

Both fibrin and fibrinogen may interact with αIIbβ3, each acting through distinct epitopes [40]. Fibrinogen binds to αIIbβ3 via the carboxy-terminal peptide sequence of the γC-peptide (GAKQAGDV), while fibrin binds to the integrin through a unique sequence in the γC-peptide, ATWKTRWYSMKK, which binds to the αIIb β-propeller [77]. It was also noted that platelet activation through αIIbβ3 is required for expression of both immobilized D-dimer (iD-dimer) and fibrinogen binding [78].

Platelets also express glycoprotein VI (GPVI). GPVI is the main platelet receptor for collagen, which is exposed during endothelial damage or dysfunction. GPVI may be found in two states, monomeric and dimeric GPVI [48,78,94]. Collagen and other substrates bind monomeric GPVI, which induces its dimerization with an adjacent GPVI [95]. Dimerization of GPVI is required for collagen binding and initiation of signalling through the associated FcR-γ chain [85]. It was found by [78] that only dimeric GPVI can interact with fibrinogen D-domain at a site proximate to its collagen binding site to support platelet adhesion/activation/aggregate formation on immobilized fibrinogen and polymerized fibrin. However, contrasting observations have been reported on whether fibrin(ogen) binds to monomeric or dimeric GPVI or to neither forms [85]. Both fibrinogen [76,78] and fibrin [48,96] can interact with GPVI on platelets [97].

Figure 5 shows some of the relevant signalling pathways where fibrin(ogen) and D-dimer are involved in platelet activation [76,79,86,87]. These signalling events initially cause platelet activation and aggregation and conformationally shape change, clot formation and eventually clot retraction [40,79,86,87,90,98]. The increase in platelet activation and aggregation during COVID-19 is therefore the result of increased mitogen-activated protein kinase (MAPK) pathway activation [99].

#### 2.1.3. Fibrin(Ogen) Binding to Endothelial Cells

Fibrinogen binds to endothelial cells (ECs) via the interactions with “intracellular” adhesion molecule 1 (ICAM-1), integrin α_V_β_1_ and integrin α_V_β_3_ [100]. ICAM-1 is an immunoglobulin (Ig)-like adhesion molecule expressed on the membrane of leukocytes and endothelial cells. Extracellularly, ICAM-1 consists of five Ig-like domains, which are mostly hydrophobic, and form β-sheets when folded. ICAM-1 has short transmembrane and cytoplasmic regions (24 and 28 amino acids, respectively) [75]. Activation of ICAM-1 leads to EC structure changes, creating gaps between ECs [70,72,75,100]. These gaps are thought to facilitate the final steps of leukocyte transmigration [75]. Leucocyte transmigration is an important occurrence in the inflammatory response and involves the recruitment of blood leukocytes to a site of injury or infection, resulting in leukocyte adhesion to the endothelial lining, diapedesis (the passage of blood cells through the intact walls of the capillaries) or transmigration across the endothelial monolayer followed by directed migration to a site of infection or injury that often involves transmigration across epithelia [101].

During inflammation, fibrin(ogen) contributes to the transmigration of leukocytes from blood vessels to inflamed tissues. This is achieved by simultaneously binding to ICAM-1 and integrin α_M_β_2_ on ECs and leukocytes [70]. However, during hyperfibrinogenaemia, fibrinogen contributes to endothelial dysfunction, proliferation and angiogenesis [66,70,71,72,73,74,75,100]. Endothelial dysfunction also increases the risk of thrombus formation [7,8,9,10,11,12,13,14,15,16,17,18,19,20,21,22,23,24,25,26,27,28,29,30,31,32,33,34,35,36,37,38,39,40,41,42,43,44,45,46,47,48,49,50,51,52,53,54,55,56,57,58,59,60,61,62,63,64,65,66,67,68,69,70,71,72,75,100]. Vasoconstriction was also shown to be mediated by fibrinogen and fragment D (early degradation product of fibrin(ogen)), when they bind to vascular ICAM-1 [47]. This might suggest that iD-dimer, too, may bind to ICAM-1 and that it is, in part, responsible for increased vascular tone and resistance, which compromises blood circulation. Figure 6 depicts some of the main intracellular pathways elicited upon fibrinogen (and possibly iD-dimer) binding to their receptors on endothelial cells.

#### 2.1.4. Fibrin(Ogen) and D-Dimer Binding to Erythrocytes

Fibrinogen is of course a major component of the coagulation cascade as well as a significant determinant contributing to plasma viscosity and ESR levels [83]. Fibrin(ogen) is also considered the main plasma protein responsible for increased erythrocyte shear-dependent reversible aggregation, contributing largely to vessel occlusion [39]. Fibrin(ogen) is thought to serve as a bridging molecule between erythrocytes during aggregation [39]. The transient bridging of two erythrocytes, promoting erythrocyte aggregation, can represent an important cardiovascular risk factor [103]. Although a search on Pubmed did not show any publications yet on the presence of Rouleau formation in Covid-19 infection, it would be an interesting phenomenon to investigate.

Fibrinogen may also play an important role in erythrocyte deformability [104]. Fibrinogen might interact with erythrocytes via integrin-like receptors; however, there are no consensus on the presence of such receptor [83]. Carvalho and co-workers in 2011, [82] suggested that a α_IIb_β_3_-like receptor might be present on erythrocytes; also, [47] argued that there might be such an integrin-like receptor. However, [83] could not find any evidence of such an integrin-like receptor and rather suggested that CD47 is a fibrinogen ligand on erythrocytes, and CD47 expression was found to be decreased on the surface of erythrocytes in obese individuals [84]. The authors suggested that changes in CD47 expression on the erythrocyte surface may be an adaptive response to hyperfibrinogenemia associated with obesity [84]. iD-dimer might also possibly bind to such an integrin-like receptor or CD47 on erythrocytes. It was also hypothesized that erythrocyte membrane damage during COVID-19 due to binding of inflammatory molecules may result in critical biophysical events, like bubble nucleation or foaming [105]. In addition to directly binding to erythrocytes, fibrin(ogen) influences erythrocyte functionality by increasing circulating inflammatory biomarkers by binding to ECs. The presence of inflammatory biomarkers in circulation are associated with reactive oxygen species (ROS) production, which cause erythrocyte eryptosis and pathological deformability [106,107]. The increased viscosity of blood due to hyperfibrinogenaemia may also increase shear flow rates [71]. This, along with inflammation, leads to a phosphatidyl serine (PS) flip on the erythrocyte membrane. The exposure of PS on erythrocytes is known to be present during pathological coagulation and can in turn be involved in the production of thrombin [39]. Under pathological conditions, such as chronic inflammation, the PS flip contributes to increased erythrocyte aggregation [39,106]. PS also mediates the adhesion of erythrocytes to vessel walls, promoting occlusion of small vessels [39]. During COVID-19 infection, pathological levels of thrombin, fibrin(ogen), D-dimer and increased circulating inflammatory molecules may interact with erythrocytes, resulting in fragile erythrocyte membranes, with pathological elasticity. These erythrocytes may in turn be trapped in embolisms and clots formed in COVID-19 patients. Normal (to slightly increased) levels of D-dimer are noted early in the progression of the disease; however, as the disease progresses, D-dimer levels increase rapidly (clinical observation).

### 2.2. The Importance of Von Willebrand Factor (VWF) as a Circulating Biomarker

Von Willebrand factor (VWF) is a multimeric glycoprotein present in plasma and the subendothelial matrix [108]. It is stored in the form of ultra-large (UL) VWF multimers in Weibel-Palade bodies and platelet α-granules for secretion upon stimulation. In response to high shear stress and other inflammatory mediators [109] resting ECs are activated and release large amounts of long VWF multimers into circulation. These VWF multimers are cleaved, and can be activated by the metalloprotease ADAMTS-13 [110]. Therefore, after ADAMTS-13 activation, VWF will now have an exposed binding site for GPIbα (which is part of the GPIb-IX-V receptor complex) [110]. ADAMTS-13 is produced in the liver, and its main function is to cleave VWF anchored on the endothelial surface and in circulation [111]. Subsequently, platelets bind to these activated UL-VWF string via GpIbα interaction with the exposed A1 domain (in the GPIb-IX-V complex), initiating the thrombogenic process which is summarized in Figure 7 and details of the pathway and receptors are discussed in the next section.

The production of VWF is exclusive to endothelial cells and megakaryocytes [112]. VWF is involved in platelet aggregation and thrombus formation and due to its important role in inflammation, VWF is identified as an acute phase reactant [113]. In addition, VWF is identified as a crucial player in the propagation of atherosclerosis by promoting plaque formation and inflammation [114]. This is important because atherosclerotic lesions cause obstruction, which further promotes thrombus formation (and embolization), resulting in reduced cerebral blood flow (cerebral ischemia) [115]. In addition, VWF tethers circulating platelets to the endothelium as part of the processes of coagulation, inflammation, and also tumor progression [115]. VWF also acts as a carrier—and stabilizer—of the procoagulant factor VIII (FVIII) in circulation [116] which is achieved by the formation of a non-covalently bound VWF-FVIII complex that protects FVIII from being degraded by activated protein C [117]. A most important consideration for COVID-19 pathology, is that, under normal conditions, VWF is both a bleeding (when low) and thrombotic marker (when raised) [109]. However, the stage of the disease is of significant importance for treatment (see also our discussion in the Conclusion section).

#### 2.2.1. Von Willebrand Factor (VWF) Receptors and Pathways on Platelets

VWF binds to two distinct platelet receptors which are localized on the platelet membrane; they are GPIbα in the GPIb-IX-V complex [112,118] and integrin αIIbβ3 (GPIIb-IIIa complex) [112]. In previous sections, we discussed α_II_bβ3 in detail. The GPIb-IX-V complex comprises of 2 chains of GPIbα (135kDa), 2 GPIbβ (26 kDa), 2 GPIX (20 kDa) and 1 GPV (82kDa) at a ratio of 2:2:2:1. All 4 proteins belong to the leucine-rich repeat (LRR) superfamily [119]. Binding of VWF to GPIbα causes activation of tyrosine protein kinases LYN and FYN, which are members of the Src family kinase (known as non-receptor tyrosine kinases). Activation of these kinases leads to tyrosine phosphorylation of the immunoreceptor tyrosine-based activation motif (ITAM) on the FcRγ or FcγRIIa receptors with which GPIb physically associates [119].

Engagement of GPIb-IX-V activates intracellular signalling events that lead to full platelet activation and aggregation through the α_II_bβ3 integrin [119]. The binding of VWF to GPIb-IX-V leads to the upregulation of α_II_bβ3 integrin affinity [120]. VWF can then bind to α_II_bβ3, thus enhancing platelet adhesion and platelet aggregation and contributing to thrombus formation by binding to fibrinogen [120], which is mediated via different pathways. Intracellular signalling that induces changes in the extracellular ligand-binding domain of integrins from a low-affinity state to the activated or high-affinity resulting inside-out signalling [121]. Figure 7 shows some of the signalling pathways of VWF.

#### 2.2.2. Von Willebrand Factor Receptors and Pathways on Endothelial Cells

Although the pivotal physiological role of VWF is to activate platelets, binding to ECs has been demonstrated. αvβ3 is the major integrin expressed on ECs [122]. Although αvβ3 binds multiple ligands such as vitronectin, fibrinogen and fibronectin, αvβ3 is the best-characterized EC receptor for VWF [122]. The most extensively studied function of αvβ3 in vascular biology relates to endothelial cell (and smooth muscle cell) adhesion, migration proliferation, differentiation and survival [116]. The complex responses which rely on these functions of αvβ3 include angiogenesis, vasculogenesis and vascular cell survival [122,123]. It has been suggested that this αvβ3 integrin may be central in the inflammatory endothelial responses [123]. However, very little is known about the signalling events that follow VWF binding to αvβ3 on EC [124].

#### 2.2.3. Von Willebrand Factor (VWF) Signalling in Erythrocytes

VWF can bind to erythrocytes under conditions such as reduced shear rates [125]. The erythrocyte surface receptor(s) subdomains in VWF that mediate this adhesion are unknown [126]. Upon an inflammatory insult and increased production of reactive oxygen species (ROS), ECs are prompted to release VWF from the Weibel–Palade bodies, resulting in elevated VWF levels observed in an inflammatory state [127]. Following secretion from Weibel–Palade bodies, a portion of VWF enters into circulation while another portion remains bound to the endothelial surface [128]. During inflammation (and oxidative stress), erythrocytes can also undergo eryptosis [106]. Eryptosis is characterized by three distinct physiological processes which are cell shrinkage, membrane blebbing and cell membrane scrambling [106,129]. Subsequent to membrane scrambling, phosphatidylserine (PS) translocates from the inner leaflet of the cell membrane and is exposed on the erythrocyte surface [130]. Nicolay et al. [131] showed that the exposure of PS and Annexin V, which avidly binds to PS, mediates binding of eryptotic red blood cells (RBCs) to VWF. The A1 domain of VWF is mainly responsible for mediating this VWF–erythrocyte adhesion [131]. Thus, VWF promotes erythrocyte–erythrocyte linking [131]. Moreover, intraluminal VWF mediates platelet-independent erythrocyte adhesion to ECs, thus mediating microvascular occlusion and impaired dynamic blood flow [125].

### 2.3. The Importance of P-Selectin as a Circulating Biomarker

P-selectin, also known as CD62P, play an important role in modulating interactions between blood cells and endothelial cells [132]. P-selectin is constitutively present in α-granules of platelets and Weibel–Palade bodies in endothelial cells [132,133,134]. P-selectin also found in human plasma—here P-selectin—is alternatively spliced and lacks the transmembrane domain [134]. This is referred to as soluble P-selectin (sP-selectin) [132]. As membrane receptor, P-selectin acts as an adhesion receptor to support leukocyte rolling and emigration at sites of inflammation [134]. Figure 8 shows a simplified overview of P-selectin interactions with platelets and neutrophils.

Since P-selectin is stored in and expressed by endothelial cells and platelets, there has been substantial debate on whether raised plasma levels of P-selectin indicate endothelial dysfunction, platelet activation or both [132]. Elevated levels of sP-selectin may also reflect platelet activation, since P-selectin is proteolytically shed from the plasma membrane in vivo shortly after activation [40]. Plasma levels of sP-selectin have also been considered a useful biomarker in cardiovascular diseases since sP-selectin is constantly elevated in such patients [136]. Increased sP-selectin can therefore also reflect endothelial cell activation and damage [137]. Pathological levels of sP-selectin consistently promote leukocytes to adhere to endothelial via the activation of the leukocyte integrin Mac-1 [138]. Circulating sP-selectin is also thought to trigger signalling in leukocytes that has a direct contribution to inflammation and thrombosis. However, sP-selectin likely circulates as a monomer, and in vitro studies propose that sP-selectin must dimerize to induce signalling in leukocytes [136]. When sP-selectin is dimerized, it can trigger activation of the leukocytes in vitro, manifest as leukocyte adhesion to ICAM-1 and to fibrinogen, and release citrullinated histones and neutrophil extracellular traps (NETs) [139].

#### 2.3.1. P-Selectin Signalling in Platelets

P-selectin glycoprotein ligand-1 (PSGL-1) is the primary receptor for P-selectin [138] and is a 120kDA transmembrane protein that is mostly expressed as a homodimer rich in *O*- as well as *N*-glycans [132,140]. P-selectin from endothelial cells also binds to the GPIbα [141], which is part of the platelet receptor complex GPIb-V-IX that promotes platelet aggregation. P-selectin plays an important role in neutrophil–platelet, platelet–platelet and monocyte–platelet interactions [141]. P-selectin on activated platelets in suspension can also bind to PSGL-1 on neutrophils or monocytes, contributing to the formation of mixed cell aggregates [142]. When platelets are exposed to agents such as adenosine or epinephrine, the platelet can become activated and there is an increase of P-selectin expression on the platelet surface [132]. Thus, P-selectin is the most important signal molecule during pathological coagulation as well as infection.

#### 2.3.2. P-Selectin Signalling in Endothelial Cells

Early inflammatory mediators like histamine, thrombin, hypoxia or phorbol esters can stimulate EC in vitro, causing endothelial damage or endotheliopathy. These agonists mobilize P-selectin to the apical membranes of EC where P-selectin initiates the rolling adhesion of flowing neutrophils [143]. Although not the focus of this review, the interactions of the rolling of neutrophils, P-selectin and the endothelial cells are shown in a simplified diagram, Figure 9. P-selectin in Weibel–Palade bodies are mobilized in a process of degranulation [132]. P-selectin has at least two waves of aggregation at the cell surface: one at 10 min and the other at 12 h after endotoxic or oxidative stress [144]. In addition, neutrophils rolling on P-selectin secrete the cytokine oncostatin M. The released oncostatin M can also trigger signals through glycoprotein 130 (gp 130)-containing receptors on ECs that result in a further clustered P-selectin and markedly enhanced adhesive function [143]. P-selectin also interacts with platelet sulfatides, thereby stabilizing initial platelet aggregates formed by GPIIb/IIIa–fibrinogen bridges [145]. Oxidatively modified fibrinogen can also cause platelet aggregation and potentiates ADP-induced platelet aggregation and production of active oxygen forms in zymosan-stimulated leukocytes [146]. The oxidized form of fibrinogen impairs microrheological properties of the blood, significantly reduces erythrocyte deformability, increases blood viscosity and reduces suspension stability of the blood [146].

#### 2.3.3. P-Selectin Interaction with Erythrocytes

Erythrocytes are not considered to participate in the receptor-mediated processes seen in endothelial cells and platelets, since normal red blood cells (RBCs) are not known to bear selectin ligands or to bind to P-selectin [151]. However, studies focused on sickle cell disease have indicated that the adhesion of sickle erythrocytes to the vascular endothelium may be potentiated by the upregulation of adhesion molecules on activated endothelial cells [151]. Normal and, to a greater extent, sickle erythrocytes adhere to endothelial P-selectin. The adherence of sickle erythrocytes is problematic since this adhesion may contribute to vaso-occlusion [151]. The binding of sickle cells is much higher than normal cells, since in sickle cell disease there are multiple adhesion systems involved [151]. Abnormal adhesion of erythrocytes to the endothelial layer is linked to the pathophysiology of various vascular disorders [152]. Abnormal adhesion is a result of several biochemical changes in erythrocyte membranes and may include exposure of PS on erythrocytes outer membranes and plasma protein levels [152]. Adhesion between EC, erythrocytes and fibrin(ogen) plays an important role in the hyperactivation of the coagulation system during inflammation.

When P-selectin is upregulated in circulation, endotheliopathy as well as platelet hyperactivation may be prevalent. In addition, P-selectin plays a fundamental role in adhesion of erythrocytes to damaged endothelia as well as to adjacent erythrocytes and to hyperactivated platelets. Recently, it was reported that endotheliopathy is present in patients with COVID-19 and that the presence of endothiopathy is likely to be associated with critical illness and death [153]. The authors came to this conclusion after studying endothelial cell damage, levels of platelet activation, VWF antigen, soluble thrombomodulin, soluble P-selectin and soluble CD40 ligands, various coagulation factors, endogenous anticoagulants and fibrinolytic enzymes [153].

## 3. Conclusions

It is very well known that pathological levels (both decreased and increased levels) of fibrin(ogen), D-dimer, VWF and P-selectin play crucial roles in abnormal coagulation and endothelial dysfunction. These molecules may also be significantly dysregulated in patients with COVID-19, as reviewed in the introduction. Specifically, dysregulation during COVID-19 has been noted in P-selectin [8], fibrinogen and D-dimers [4,5,9,10,11,12,13,14,15], and VWF [16,17]. Depending on the direction, dysregulation of fibrin(ogen) D-dimer, VWF and P-selectin may result in either hypercoagulation or excessive bleeding and thrombocytopenia. During typical bleeding and abnormal clotting diseases, low fibrinogen levels are indicative of a higher propensity for bleeding while high levels are known to be associated with hypercoagulation [154]. We summarized these phenomena in Figure 2A. Bleeding as well as thrombotic events often occur in subjects with multiple weak risk factors which interact to produce the symptoms [109]. Both bleeding, thrombocytopenia and thrombotic pathologies have been reported in COVID-19 patients and are significant accompaniments to acute respiratory distress syndrome and lung complications [3,4,5,6,12,155,156]. Therefore, fibrin(ogen) levels, D-dimer, VWF and P-selectin could all be valuable biomarkers that might provide clinicians with the correct clinical diagnosis and might assist in deciding the method of treatment.

Our understanding of this process is explained in Figure 2B, and clinical observation suggests the following: during early-stage COVID-19, in patients presenting with normal to slightly increased D-dimer levels, increased levels of fibrinogen, VWF and P-selectin and slightly activated platelets, if untreated, the clinical picture changes to a rapid increase in D-dimer; still higher levels of fibrinogen, VWF and P-selectin; and hyperactivation of platelets. This is in line with hyperclotting or thrombosis. In the critically ill patients, D-dimer and P-selectin levels are high while fibrinogen and VWF levels are decreased as these molecules are depleted from either circulation or the damaged endothelial cells and hyperactivated platelets that now show thrombocytopenia. During these late stages of the progression of the disease, the cytokine storm is also prevalent.

D-dimer assays to determine the levels of D-dimer in circulation are also very helpful, as is an indirect marker of fibrinolysis and fibrin turnover [157]. However, in COVID-19 patients, D-dimer levels are normal or slightly increased during the early stages of the disease (clinical observation). Increased levels of circulating P-selectin are associated with a higher risk of a thrombotic event [158]. However, P-selectin expression on platelets may also be used to diagnose mild bleeding disorders and increased bleeding might be associated with very suppressed P-selectin expression [159]. A dilemma is that P-selectin expression varies considerably between individuals.

If the VWF level is increased, it predicts a thrombotic phenotype, and when these levels are low in plasma, the phenotype is indicative of bleeding [109]. Thrombotic risk may be more prevalent when VWF is activated and increased and may be able to bind to the various receptor complexes (Figure 7). Heparin inhibits VWF-GP1b binding, and the reason for this is because heparin overlaps the binding site within the VWF A1 domain [109]. Heparin has also been found in some circumstances to be a helpful treatment for COVID-19 [160,161]. Heparin interferes with VWF platelet activation, and possibly assists in the prevention of thrombotic events. When the bleeding phenotype is more prevalent, it may be due to abnormalities in subendothelial collagen, which may alter its interaction with platelets and VWF [162]. However, if VWF is depleted, it results in bleeding. During the late stages of COVID-19, VWF levels are indicative of depletion (clinical observation by co-author Laubscher) and this is due to large-scale endothelial damage. Endotheliopathy is also prevalent in patients with COVID-19 [153] and is significantly linked to coagulopathies and clinical observations.

Personalised medicine has never been so important as during this epidemic. This is mainly because the patient cohort is so extremely diverse and because many may have preexisting thrombotic disease and cardiovascular comorbidities [155]. Point-of-care devices and diagnostics like the thromboelastograph (TEG) (that gives an indication of fibrinogen levels) or point-of-care ultrasound (POCUS) [163] allows for frequent testing of the coagulation/bleeding profiles as well as blood clot fibrinolysis of patients at the bedside [163]. TEG is particularly useful to also assess fibrinolysis, in COVID-19 patients [156]. The TEG can also predict thromboembolic events in critically ill patients with COVID-19 [156].

Most importantly, we need clinicians to have access to adequate point-of-care devices and encourage them to use parameters of haemostasis [10], including fibrinogen D-dimer and VWF levels, to allow them to determine if their patients are in need of either therapeutic antithrombotic prophylaxis/treatment or fibrinolytic therapy to prevent whichever coagulopathy is present, be it hypercoagulation, fibrinolysis shutdown or bleeding. Most significant is that patients need to be treated early in the disease progression when high levels of VWF, P-selectin and fibrinogen are present with still low levels of D-dimer. Progression to VWF and fibrinogen depletion with high D-dimer levels and even higher P-selectin levels, followed by the cytokine storm, will be indicative of a poor prognosis.

## Figures and Tables

**Figure 1 ijms-21-05168-f001:**
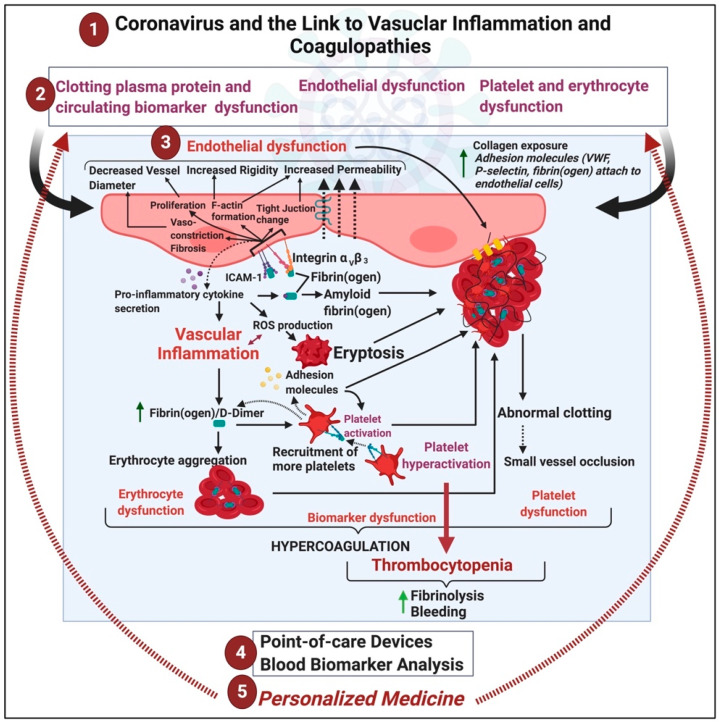
(**1**) Vascular implications of acute respiratory syndrome coronavirus 2 (COVID-19) (**2**) may result in clotting protein and circulating biomarker, endothelial, and erythrocyte and platelet dysfunctions. (**3**) We review the various biochemical processes associated with vascular dysfunction, focussing on fibrin(ogen), D-dimer, P-selectin and von Willebrand Factor. (**4**) We conclude by looking at point-of-care devices and methodologies in COVID-19 treatment and suggest that each patient should be treated using a (**5**) personalized medicine approach. This image was created with BioRender (https://biorender.com/).

**Figure 2 ijms-21-05168-f002:**
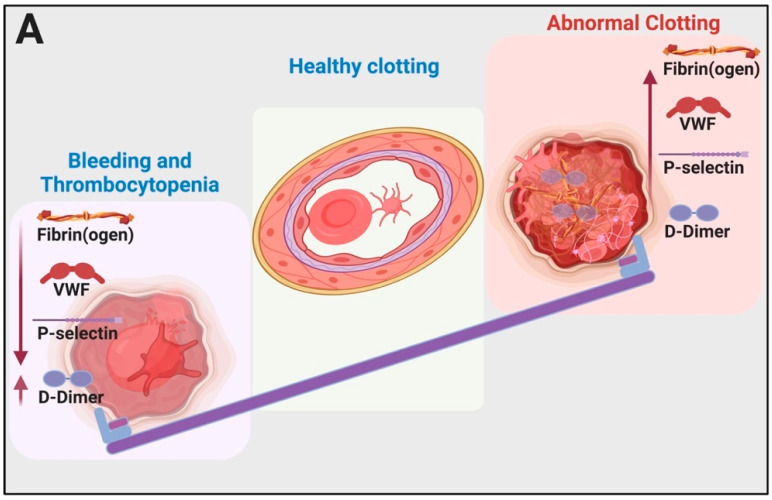
(**A**) Typical pathology in bleeding and clotting: the seesaw balancing act between bleeding and thrombocytopenia and hypercoagulation. This image was created with BioRender (https://biorender.com/). (**B**) Clinical manifestation of hypercoagulation, thrombocytopenia and bleeding during COVID-19 as well as clinical care options and optimal time for intervention: During the early stages of abnormal clotting, D-dimer levels are normal or slightly increased but will increase rapidly with progression of the disease (clinical observation). This image was created with BioRender (https://biorender.com/).

**Figure 3 ijms-21-05168-f003:**
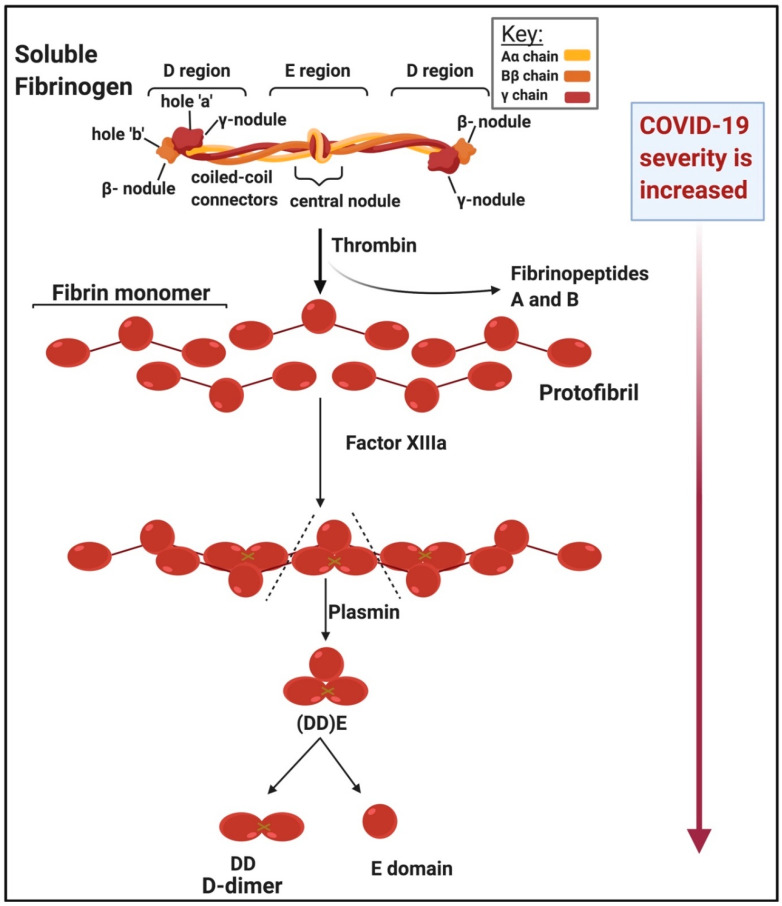
Structure of soluble versus insoluble fibrin(ogen) and the action of thrombin and D-dimer formation adapted from [41]: D-dimer levels are increased in very ill patients (clinical observation). This image was created with BioRender (https://biorender.com/).

**Figure 4 ijms-21-05168-f004:**
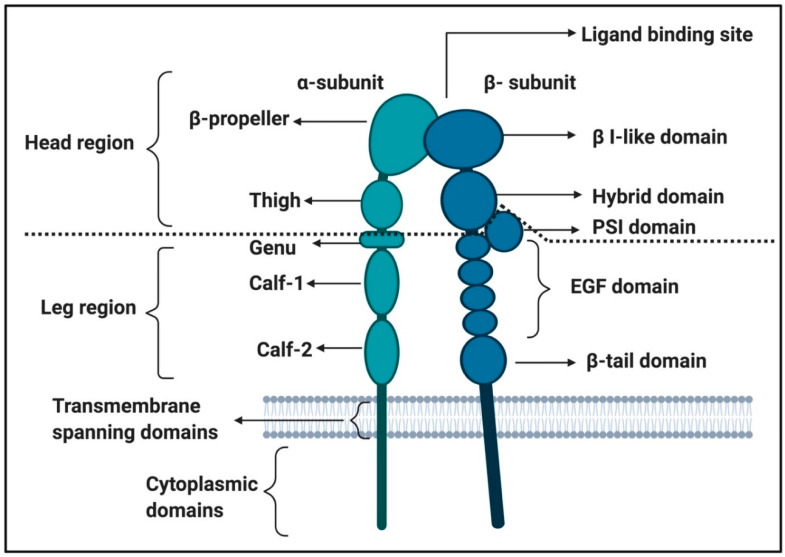
Integrin αIIbβ3 structure adapted from [79,89,90,91]: This figure was created using BioRender (https://biorender.com/).

**Figure 5 ijms-21-05168-f005:**
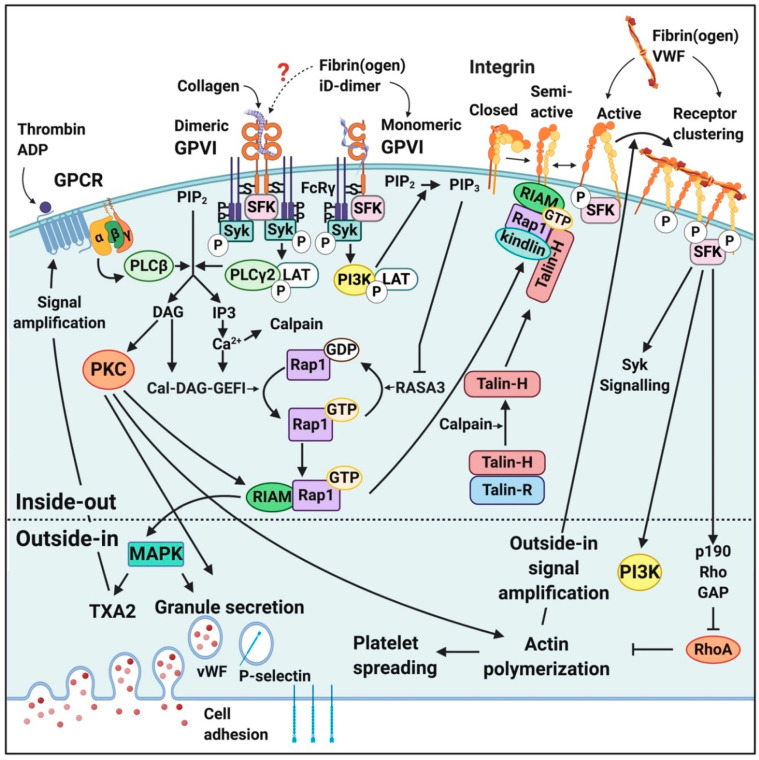
Activation of a platelet, inside-out and outside-in signalling upon ligation of major platelet membrane receptors. Adapted from [40,76,86,90,98]. Abbreviations: GPCR, G-protein coupled receptor; GPVI, glycoprotein VI; VWF, von Willebrand’s factor; PIP2, phosphatidylinositol 4,5-bisphosphate; PIP3, phosphatidylinositol (3,4,5)-trisphosphate; SFK, Scr family kinases; Syk, spleen tyrosine kinase; PLC, phospholipase C; LAT, linker for activation of T-cells; DAG, diacylglycerol; IP3, inositol triphosphate; PKC, protein kinase C; Ca^2+^, calcium ions; Cal-DAG-GEFI, diacylglycerol regulated guanine nucleotide exchange factor I; GDP, guanine diphosphate; GTP; guanine triphosphate; RASA3, Ras GTPase-activating protein 3; RIAM, Rap1-GTP interacting adapter molecule; MAPK, mitogen-activated protein kinase; TXA2, thromboxane A2; GAP, GTPase activating protein; PI3K, phosphatidylinositide 3-kinase. This image was made using BioRender (https://biorender.com/).

**Figure 6 ijms-21-05168-f006:**
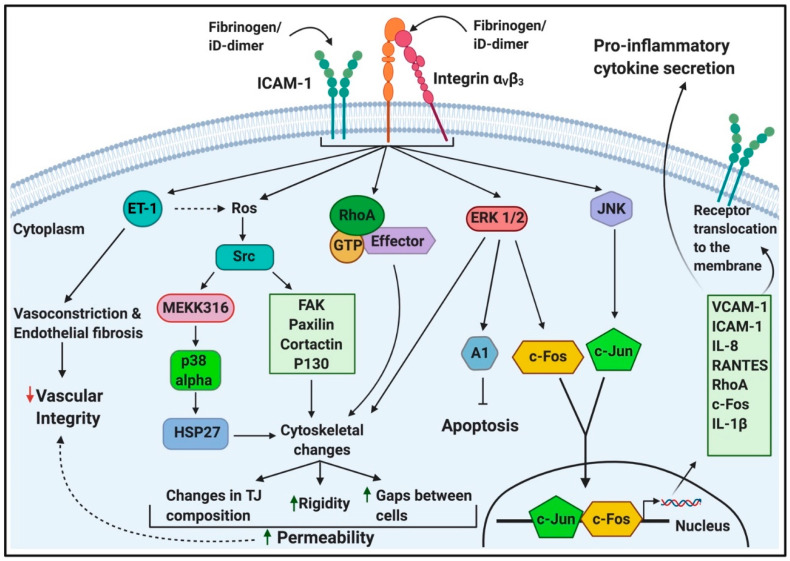
The intracellular signalling of endothelial cells upon ligation of fibrinogen (and iD-dimer): adapted from [70,71,72,73,75,100,102]. Abbreviations: ICAM-1, intercellular adhesion molecule-1; RhoA, activated Rho factor protein; iD-dimer, immobilised D-dimer; ROS, reactive oxygen species; Src, sarcoma; MEKK316, endothelial MAP kinases 3 and 6 (MKK-3 and -6); FAK, focal adhesion kinase; p130, retinoblastoma-like protein-2; p38, mitogen-activated protein kinase; HSP27, heat shock protein 27; ERK, extracellular signal-regulated kinase/extracellular receptor kinase; c-Fos, proto-oncogene; JNK, stress-activated protein kinase; c-Jun, protein encoded by the JUN gene; VCAM-1, vascular cell adhesion protein-1; IL-8, interleukin-8; RANTES, regulated upon activation normal T cell expressed and presumably secreted; IL-1 beta, interleukin-1 beta. This image was made with BioRender (https://biorender.com/).

**Figure 7 ijms-21-05168-f007:**
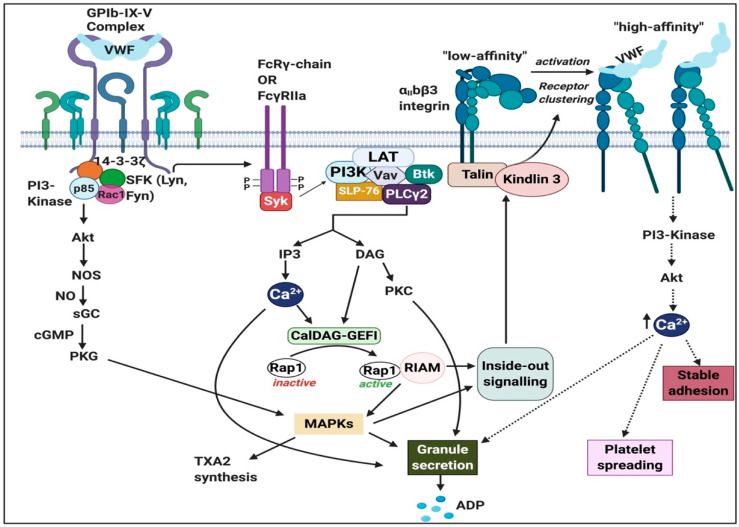
The signalling pathway of von Willebrand factor (VWF) in platelets via the GPIb-IX-V complex (with GPIbα) leading to integrin α_II_bβ3 activation and the important role of the FcRγ-chain and FcγRIIa immunoreceptor tyrosine-based activation motif (ITAM) pathway: Adapted from [112,118]. Abbreviations: ADP, adenosine triphosphate; Btk, Bruton tyrosine kinase; CalDAG-GEFI, Ca^2+^-dependent guanine nucleotide exchange factor; DAG, diacylglycerol; IP3, inositol triphosphate; LAT, linker of activated T-cells; MAPK, mitogen-activated protein kinase; NO, nitric oxide; NOS, nitric oxide synthase; PI3K, phosphoinositide 3; PLC-γ_2,_ phospholipase C-γ_2_, PKC, protein kinase C; PKG, protein kinase G; Rac 1, Ras-related C3 botulinum toxin substrate 1; Rap1, Ras-related protein 1; RIAM, Rap1-GTP-interacting adaptor molecule; SFK, Src family kinases; SLP-76, SH2 domain-containing leukocyte phosphorylation of 76 kDa; SYK, spleen tyrosine kinase; sGC, soluble guanine cyclase; TXA2, thromboxane A2; VWF, von Willebrand factor. Figure created using Biorender (https://biorender.com/).

**Figure 8 ijms-21-05168-f008:**
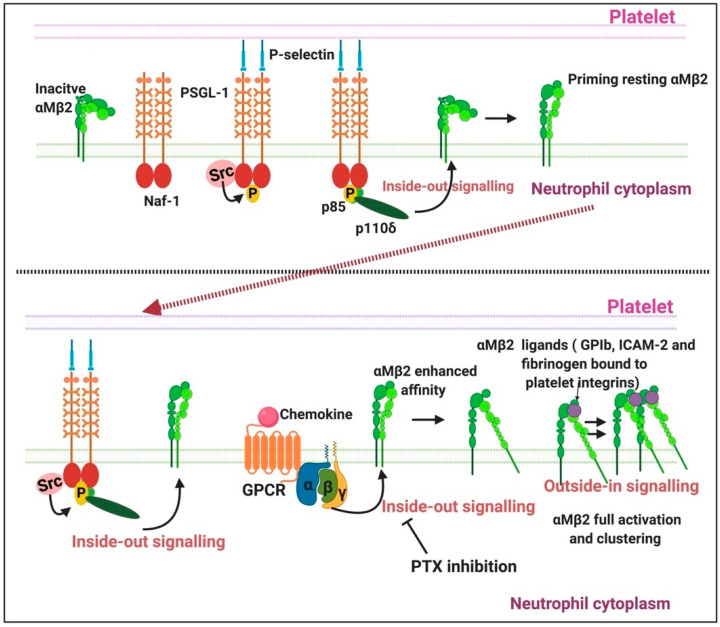
The signalling pathways involved in neutrophil and platelet activation: Adapted from [135]. Abbreviations: GPCR: G protein-coupled receptor; Mac-1 or (αMβ2), macrophage antigen-1; P-sel, P-selectin; PSGL-1, P-selectin glycoprotein ligand-1; PTX, pertussis toxin. Figure created using BioRender (https://biorender.com/).

**Figure 9 ijms-21-05168-f009:**
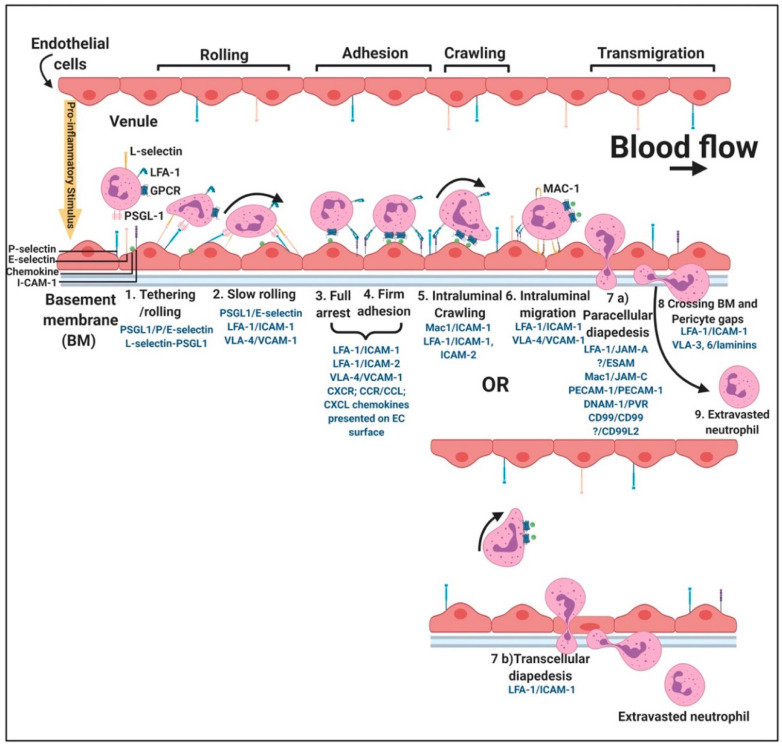
Simplified leukocyte extravasation cascade and rolling of neutrophils over the endothelium: adjusted from [102,147,148,149,150]. For more detail, under each step, the known adhesion receptor interactions are given with the leukocyte receptor being named first. Unknown ligands are indicated with question marks. Abbreviations: GPCR- G-protein coupled receptor, LFA-1, lymphocyte function-associated antigen 1; PSGL-1, P-selectin glycoprotein ligand-1; I-CAM-1, intercellular adhesion molecule 1. Figure created using BioRender (https://biorender.com/).

**Table 1 ijms-21-05168-t001:** Receptors known to bind fibrinogen and D-dimer and the effects they elicit within different cell types.

Cell Type	Receptor	Effect	References
Endothelial cells (EC)	Integrins α_V_β_3_ and α_5_β_1_	Endothelial cell proliferation, endothelial cell activation, angiogenesis, increased EC permeability and vasoconstriction	[34,66,67,68,69]
	Integrin α_M_β_2_	Facilitates interaction fibrinogen with ICAM-1 during leukocyte transmigration	[70]
	ICAM-1	Platelet adhesion, leucocyte adhesion and transmigration to site of infection, mitogenesis, angiogenesis, cell survival, release of pro-inflammatory cytokines, ICAM-1 receptor recruitment to EC membrane and vasoconstriction	[70,71,72,73,74,75]
Platelets	Glycoprotein IIb/IIIa	Platelet activation and spreading, integrin activation and granule secretion	[71]
	Glycoprotein VI (GPVI)	Platelet activation and spreading, integrin activation and granule secretion	[40,48,76,77,78]
	Integrins α_IIb_β_3_ α_V_β_3_	Outside-in signalling in platelets—platelet spreading and granule secretion	[34,40,67,79,80,81]
Erythrocytes	αIIbβ3-related type integrin? CD47	Erythrocyte aggregation and adhesion	[47,82,83,84]
iD-dimer and/or Fragment D			
Endothelial cells	ICAM-1	Arterial constriction	[47]
Platelets	Integrin αIIbβ3	Platelet spreading and aggregation	[78]
	GPVI (monomeric/dimeric?)	Platelet spreading	[48,85]

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
