# Peer review of "Covid-19: The Rollercoaster of Fibrin(Ogen), D-Dimer, Von Willebrand Factor, P-Selectin and Their Interactions with Endothelial Cells, Platelets and Erythrocytes"

_ijms, 2020, doi:10.3390/ijms21145168_

Round 1

Reviewer 1 Report

In this paper, the authors review the interaction of various physiological pathways that may be affected by COVID-19, but focusing primarily on fibrin(ogen), P-selectin, D-dimers, and VWF. I have some suggestions for improvement:

  1. The current recommended abbreviation for von Willebrand factor is VWF (not vWF). Please change throughout, including figures and tables.
  2. There is some duplication of text that should be deleted; some examples: (a) page 2, line 58-59: “The normal range of D-dimer is <0.50 μg/mL” repeated at line 94-95. Also, specify that this is in Fibrinogen Equivalent Units (FEU). (b) Page 3, line108: “thrombocytopenia in COVID-19 is also well recognized [26-28].” Is repeated with a separate ref on line 116. (c) page 2, lines 61-62: “fibrinogen (normal levels between 2–4 mg/mL)” repeated on page 6, lines 150-151.
  3. Page 2, lines 46-48: “This is entirely consistent with infection-induced inflammatory changes as observed in patients with disseminated intravascular coagulopathy”. Well, as the authors go on to describe in some detail, the pathology associated with COVID-19 is not really “…consistent with infection-induced inflammatory changes as observed in patients with disseminated intravascular coagulopathy”. Indeed, the only marker consistent with DIC in early stages of the disease is D-dimer; other tests re more reflective of activation/acute phase changes – until such time as the disease really progresses to terminal stages, where DIC like changes are more apparent.
  4. Page 2: discussion on ESR. I can understand the authors desire to include this discussion, where most others have focused on other acute phase changes such as CRP, VWF, FVIII, etc, but ESR is perhaps less specific a test, and most ‘sick people’ will have elevated ESR. Also, line 72: “… may lead to bleeding”. Perhaps “…may be associated with bleeding”? Is there any direct evidence that low ESR leads to bleeding in COVID-19 or otherwise?
  5. Figure 2: The figure shows abnormal clotting associated with raised fibrinogen, VWF, P-selectin and D-dimer (right) vs ‘bleeding’ associated with lowered fibrinogen, VWF, P-selectin and D-dimer (left). However, D-dimer would be expectedly be raised in both situations in COVID-19.
  6. Table 1: text “(i)D-dimer and /or Fragment D” seems to be oddly placed/not associated with Table or reference?
  7. Page 9, line 211: “Kaneva, Martyanov, Morozova, Panteleev and Sveshnikova [77].”
  8. Page 10, line 229: ‘d-dimer’ (D-dimer)
  9. Page 13, line 337: “vWF is both a bleeding and thrombotic marker [103]’. True, but it is lowered VWF and raised VWF respectively.
  10. Page 13, line 337: “..the the…”
  11. The authors use the term ‘it’ sometimes; ‘it’ lacks clarity; replace throughout with the specific term; eg, page 14, line 373: ‘it’ could be anything, and requires several reads to identify what the authors mean.
  12. The review starts with some discussion in relation to COVID, and then later develops discussion around the markers of interest (primarily fibrin(ogen), P-selectin, D-dimers, and VWF); however, much of the discussion here has unclear specific relevance to COVID; suggest some inclusion of COVID-19 related relevance through the main section. Some example: page 15, lines 386-396; discussion on PS.
  13. Page 15, line 387: ‘Erythrocytes’ (erthryrocytes).
  14. Page 18: line 523: “During the late stages of COVID-19, vWF levels are indicatinve of depletion, and this is due to large scale endothelial damage.” Requires a reference; also indicative is misspelt.
  15. Page 19: lines 528-530: “Point-of-care devices and diagnostics like the thromboelastograph (TEG) or point-of-care ultrasound (POCUS) [158] allows for frequent testing of the coagulation/bleeding profiles as well as blood clot fibrinolysis of patients at the bedside [158].” Appears an odd statement here, given the focus of the review on fibrin(ogen), P-selectin, D-dimers, and VWF. TEG in particular cannot measure any of these markers other than giving an indication of fibrinogen level.

Author Response

Please find the reviewer comments below but also as an attachment.  The changes are indicated in blue in the Pdf.  The black (clean) version is in the journal format document.

Thank you for the fast review! Please find below, the changes and additions you suggested.  We included these in red in the manuscript.

Thanks!

Resia and co-authors

In this paper, the authors review the interaction of various physiological pathways that may be affected by COVID-19, but focusing primarily on fibrin(ogen), P-selectin, D-dimers, and VWF. I have some suggestions for improvement:

  1. The current recommended abbreviation for von Willebrand factor is VWF (not vWF). Please change throughout, including figures and tables.

CHANGED

  1. There is some duplication of text that should be deleted; some examples: (a) page 2, line 58-59: “The normal range of D-dimer is <0.50 μg/mL” repeated at line 94-95. Also, specify that this is in Fibrinogen Equivalent Units (FEU).

The normal range of D-dimer is < 0.50 μg/mL (Fibrinogen Equivalent Units (FEU)).

Removed repeat, sentence now reads: Furthermore, it was noted that D-dimer on admission was fourfold increase…

(b) Page 3, line108: “thrombocytopenia in COVID-19 is also well recognized [26-28].” Is repeated with a separate ref on line 116. (c) page 2, lines 61-62: “fibrinogen (normal levels between 2–4 mg/mL)” repeated on page 6, lines 150-151.

Together with fibrin(ogen) and D-dimer  analysis, thrombocytopenia in COVID-19 is also well recognized (Lippi et al., 2020, Yang et al., 2020, Xu et al., 2020, Amgalan and Othman, 2020).

Removed second referral to normal levels between 2–4 mg/mL: sentence now reads:

Fibrinogen is also an acute-phase protein that is upregulated during inflammation (Davalos and Akassoglou, 2012, Randeria et al., 2019, Weisel and Litvinov, 2017)…

  1. Page 2, lines 46-48: “This is entirely consistent with infection-induced inflammatory changes as observed in patients with disseminated intravascular coagulopathy”. Well, as the authors go on to describe in some detail, the pathology associated with COVID-19 is not really “…consistent with infection-induced inflammatory changes as observed in patients with disseminated intravascular coagulopathy”. Indeed, the only marker consistent with DIC in early stages of the disease is D-dimer; other tests re more reflective of activation/acute phase changes – until such time as the disease really progresses to terminal stages, where DIC like changes are more apparent.

Sentence was adapted to now read:

Pathology might also be consistent with infection-induced inflammatory changes as observed in patients with disseminated intravascular coagulopathy (Connors and Levy, 2020). 

  1. Page 2: discussion on ESR. I can understand the authors desire to include this discussion, where most others have focused on other acute phase changes such as CRP, VWF, FVIII, etc, but ESR is perhaps less specific a test, and most ‘sick people’ will have elevated ESR. Also, line 72: “… may lead to bleeding”. Perhaps “…may be associated with bleeding”? Is there any direct evidence that low ESR leads to bleeding in COVID-19 or otherwise?

An important marker of COVID-19 disease severity might thus also be erythrocyte sedimentation rate (ESR). An important marker of COVID-19 disease severity might thus also be erythrocyte sedimentation rate (ESR) ((Zeng et al., 2020). However, it should be noted that most patients with comorbidities will have an elevated ESR.

Reference 21 reported increased ESR levels: Zeng F, Huang Y, Guo Y, et al. Association of inflammatory markers with the severity of COVID-19: A meta-analysis. Int J Infect Dis2020; 96: 467-74.

  1. Figure 2: The figure shows abnormal clotting associated with raised fibrinogen, VWF, P-selectin and D-dimer (right) vs ‘bleeding’ associated with lowered fibrinogen, VWF, P-selectin and D-dimer (left). However, D-dimer would be expectedly be raised in both situations in COVID-19.

Interestingly, our Clinicians finds that D-Dimer levels are still in the normal levels during the early phases of the disease, the it progresses significantly when hypercoagulation and fibrin(ogen) pathology is noted.  But, perhaps the down arrow is confusing, it should be a horizontal arrow. Adjusted fig 2B

Also added the following in the Abstract:

…Most significant is that patients need to be treated early in the disease progression, when high levels of VWF, P-selectin and fibrinogen are present with still normal levels of D-dimer (however, D-dimer levels will rapidly increase as the disease progresses).

In the text:

…As mentioned previously, during early onset of the condition, D-dimer is within normal ranges, and as the patient disease severity progresses, D-dimer levels are significantly increased (clinical observation by co-author).  It was also suggested that the early evaluation and continued monitoring of D-dimer levels after hospitalization may identify patients with cardiac injury and predict further COVID-19 complications (24).  Furthermore, it was noted that, when D-dimer levels on admission was fourfold increase;  could effectively predict in-hospital mortality in patients with Covid-19 (Zhang et al., 2020)(the normal range of D-dimer is < 0.50 μg/mL). 

Table 1: text “(i)D-dimer and /or Fragment D” seems to be oddly placed/not associated with Table or reference?Changed table layout

  1. Page 9, line 211: “Kaneva, Martyanov, Morozova, Panteleev and Sveshnikova [77].”

… with other receptors (such as GPIb, GPVI, or FcγRIIa) (Kaneva et al., 2019). Page 10, line 229: ‘d-dimer’ (D-dimer) changed

  1. Page 13, line 337: “vWF is both a bleeding and thrombotic marker [103]’. True, but it is lowered VWF and raised VWF respectively.A most important consideration for COVID-19 pathology, is that, under normal conditions, VWF is both a bleeding (when low) and thrombotic marker(when raised)…
  2. Page 13, line 337: “..the the…” changed
  3. The authors use the term ‘it’ sometimes; ‘it’ lacks clarity; replace throughout with the specific term; eg, page 14, line 373: ‘it’ could be anything, and requires several reads to identify what the authors mean.

Checked paper ad clarified “it”

  1. The review starts with some discussion in relation to COVID, and then later develops discussion around the markers of interest (primarily fibrin(ogen), P-selectin, D-dimers, and VWF); however, much of the discussion here has unclear specific relevance to COVID; suggest some inclusion of COVID-19 related relevance through the main section. Some example: page 15, lines 386-396; discussion on PS.

We included the following the section where PS is described, to better link it to COVID-19:

…It was also hypothesized that erythrocyte membrane damage during COVID-19, due to binding of inflammatory molecules, may result in critical biophysical events,  like bubble nucleation or foaming (Denis, 2020).   In addition to directly binding to erythrocytes, fibrin(ogen) influences erythrocyte functionality by increasing circulating inflammatory biomarkers by binding to ECs.   The presence of inflammatory biomarkers in circulation,are associated with ROS production, which cause erythrocyte eryptosis and pathological deformability (Pretorius, 2018).….

We also included the following just before Figure 5

The increase in platelet activation and aggregation during COVID-19 is therefore  the result of  increased MAPK pathway activation (Manne et al., 2020).

  1. Page 15, line 387: ‘Erythrocytes’ (erthryrocytes). Changed

  1. Page 18: line 523: “During the late stages of COVID-19, vWF levels are indicatinve of depletion, and this is due to large scale endothelial damage.” Requires a reference; also indicative is misspelt. Changed and added (clinical observation by clinician)

However, if VWF is depleted, it results in bleeding. During the late stages of COVID-19, VWF levels are indicativeof depletion (clinical observation by co-author Laubscher)and this is due to large scale endothelial damage.  

Also added:Disseminated intravascular coagulation (DIC) in COVID-19 was also found to be accompanied by a significant decrease of fibrinogen, and a marked increase of fibrin(ogen) degradation product (FDP) formation and D-dimer (25).  Increased FDP and D-dimer are characteristics of DIC with hyperfibrinolysis, whereas the DIC caused by infection is accompanied by plasminogen activator inhibitor-1 release and suppression of fibrinolysis (25).

  1. Page 19: lines 528-530: “Point-of-care devices and diagnostics like the thromboelastograph (TEG) or point-of-care ultrasound (POCUS) [158] allows for frequent testing of the coagulation/bleeding profiles as well as blood clot fibrinolysis of patients at the bedside [158].” Appears an odd statement here, given the focus of the review on fibrin(ogen), P-selectin, D-dimers, and VWF. TEG in particular cannot measure any of these markers other than giving an indication of fibrinogen level.

This paragraph was adjusted to now read:

Point-of-care devices and diagnostics like the thromboelastograph (TEG) (that gives an indication of fibrinogen levels) or point-of-care ultrasound (POCUS) (Rubulotta et al., 2020)allows for frequent testing of the coagulation/bleeding profiles, as well as blood clot fibrinolysis of patients at the bedside (Rubulotta et al., 2020).TEG is particularly useful to also assess fibrinolysis.  In COVID-19 patients, Wright and co-workers reported fibrinolysis shutdown, because of elevated D-dimer, followed by a complete failure of clot lysis at 30 minutes on the TEG (Wright et al., 2020).   The TEG can therefore predict thromboembolic events and a need for hemodialysis in critically ill patients with COVID-19 (Wright et al., 2020).

References referred to in this document

AMGALAN, A. & OTHMAN, M. 2020. Exploring possible mechanisms for COVID-19 induced thrombocytopenia: Unanswered questions. J Thromb Haemost,18,1514-1516.

CONNORS, J. M. & LEVY, J. H. 2020. COVID-19 and its implications for thrombosis and anticoagulation. Blood,135,2033-2040.

DAVALOS, D. & AKASSOGLOU, K. 2012. Fibrinogen as a key regulator of inflammation in disease. Seminars in Immunopathology,34,43-62.

DENIS, P. A. 2020. COVID-19-related complications and decompression illness share main features.: Could the SARS-CoV2-related complications rely on blood foaming? Med Hypotheses,144,109918.

KANEVA, V. N., MARTYANOV, A. A., MOROZOVA, D. S., PANTELEEV, M. A. & SVESHNIKOVA, A. N. 2019. Platelet Integrin αIIbβ3: Mechanisms of Activation and Clustering; Involvement into the Formation of the Thrombus Heterogeneous Structure. Biochemistry (Moscow), Supplement Series A: Membrane and Cell Biology,13,97-110.

LIPPI, G., PLEBANI, M. & HENRY, B. M. 2020. Thrombocytopenia is associated with severe coronavirus disease 2019 (COVID-19) infections: A meta-analysis. Clin Chim Acta,506,145-148.

MANNE, B. K., DENORME, F., MIDDLETON, E. A., PORTIER, I., ROWLEY, J. W., STUBBEN, C. J., PETREY, A. C., TOLLEY, N. D., GUO, L., CODY, M. J., WEYRICH, A. S., YOST, C. C., RONDINA, M. T. & CAMPBELL, R. A. 2020. Platelet Gene Expression and Function in COVID-19 Patients.Blood.

PRETORIUS, E. 2018. Erythrocyte deformability and eryptosis during inflammation, and impaired blood rheology. Clin Hemorheol Microcirc,69,545-550.

RANDERIA, S. N., THOMSON, G. J. A., NELL, T. A., ROBERTS, T. & PRETORIUS, E. 2019. Inflammatory cytokines in type 2 diabetes mellitus as facilitators of hypercoagulation and abnormal clot formation. Cardiovasc Diabetol,18,72.

RUBULOTTA, F., SOLIMAN-ABOUMARIE, H., FILBEY, K., GELDNER, G., KUCK, K., GANAU, M. & HEMMERLING, T. M. 2020. Technologies to optimize the care of severe COVID-19 patients for healthcare providers challenged by limited resources. Anesth Analg.

WEISEL, J. W. & LITVINOV, R. I. 2017. Fibrin formation, structure and properties. Sub-Cellular Biochemistry,82,405-456.

WRIGHT, F. L., VOGLER, T. O., MOORE, E. E., MOORE, H. B., WOHLAUER, M. V., URBAN, S., NYDAM, T. L., MOORE, P. K. & MCINTYRE, R. C., JR. 2020. Fibrinolysis Shutdown Correlation with Thromboembolic Events in Severe COVID-19 Infection. J Am Coll Surg.

XU, P., ZHOU, Q. & XU, J. 2020. Mechanism of thrombocytopenia in COVID-19 patients. Ann Hematol,99,1205-1208.

YANG, X., YANG, Q., WANG, Y., WU, Y., XU, J., YU, Y. & SHANG, Y. 2020. Thrombocytopenia and its association with mortality in patients with COVID-19. J Thromb Haemost,18,1469-1472.

ZENG, F., HUANG, Y., GUO, Y., YIN, M., CHEN, X., XIAO, L. & DENG, G. 2020. Association of inflammatory markers with the severity of COVID-19: A meta-analysis. Int J Infect Dis,96,467-474.

ZHANG, L., YAN, X., FAN, Q., LIU, H., LIU, X., LIU, Z. & ZHANG, Z. 2020. D-dimer levels on admission to predict in-hospital mortality in patients with Covid-19. J Thromb Haemost,18,1324-1329.

Reviewer 2 Report

In this manuscript, the authors nicely reviewed the vascular implications of COVID-19 disease. Different biomarkers, including fibrin(ogen), D-dimer, vWF, and P-selectin may be changed during the progression of COVID-19. They suggested that these biomarkers may be helpful parameters for the management of COVID-19.

Overall, this is a very nice review. I have no major concerns.

Author Response

Thank you!